# SPRING: Improving the Throughput of Sharding Blockchain via Deep Reinforcement Learning Based State Placement

## Abstract

Sharding provides an opportunity to overcome inherent scalability challenges of the blockchain. In a sharding blockchain, the state and computation are partitioned into smaller groups, known as "shards," to facilitate parallel transaction processing and improve throughput. However, since the states are placed on different shards, cross-shard transactions are inevitable, which is detrimental to the performance of the sharding blockchain. Existing sharding solutions place states based on heuristic algorithms or redistribute states via graph-partitioning-based methods, which are either less effective or costly. In this paper, we present *Spring*, the first deep-reinforcement-learning(DRL)-based sharding framework for state placement. Spring formulates the state placement as a Markov Decision Process which takes into consideration the cross-shard transaction ratio and workload balancing, and employs DRL to learn the effective state placement policy. Experimental results based on real Ethereum transaction data demonstrate the superiority of Spring compared to other state placement solutions. In particular, it decreases the cross-shard transaction ratio by up to 26.63% and boosts throughput by up to 36.03%, all without unduly sacrificing the workload balance among shards. Moreover, updating the training model and making decisions takes only 0.1s and 0.002s, respectively, which shows the overhead introduced by Spring is acceptable.

**CCS Concepts:** • **Computer systems organization** → *Peer-to-peer architectures*.

*Keywords:* blockchain, sharding, scalability, deep reinforcement learning

**ACM Reference Format:**
Anonymous Authors. 2023. SPRING: Improving the Throughput of Sharding Blockchain via Deep Reinforcement Learning Based State Placement. In *Proceedings of (WWW)*. ACM, New York, NY, USA, 11 pages. https://doi.org/XXXXXXX.XXXXXXX

## 1 Introduction

Breaking the scalability trilemma [1] poses a significant challenge in blockchain technology as it requires improving throughput while maintaining a balance among safety, decentralization, and scalability. Among various scaling solutions, sharding [14, 15, 20, 27, 39, 43] stands out as a promising approach for addressing the scalability trilemma. Sharding adopts a "divide and conquer" approach by partitioning the blockchain into smaller segments called *shards*. Each shard stores a portion of the blockchain's state and can process transactions independently and concurrently, thereby enhancing the performance of the blockchain. Practically, the state in the blockchain can be represented by an *address*, a unique identifier for blockchain users to participate in transactions. Whenever a new state is stored within the blockchain, a corresponding address will be created. In this paper, we concentrate on the account-balance data model, wherein the address signifies either a user or a smart contract [40].

Although enhanced with the sharding mechanism, the throughput of sharding blockchains cannot scale linearly with an increasing number of shards, owing to the time-consuming cross-shard transactions. The *cross-shard transaction* (*CST*) is a special transaction that is used to handle the scenario where the states of transaction participants reside on different shards [24, 39, 42]. CSTs are common in sharding blockchains, with studies indicating that when there are more than 16 shards, over 95% of transactions are CSTs [24, 39]. Therefore, reducing CST is crucial to improve the throughput of sharding blockchains further.

The key to reducing CST lies in smartly placing states into different shards. However, there is a trade-off between minimizing the number of CST and maintaining a balanced workload distribution among shards. For instance, placing all states in the same shard could eliminate CST but regress to a standalone blockchain scenario. In this paper, we refer to the challenge of reducing CST while maintaining a balanced workload as the *state placement* problem.

The reduction of CSTs can be achieved in different operational phases of a sharding blockchain. As depicted in Fig.1, a sharding blockchain protocol encompasses multiple consecutive epochs. In each epoch, there are two primary phases: the *consensus phase* and the *reconfiguration phase*[15, 26].

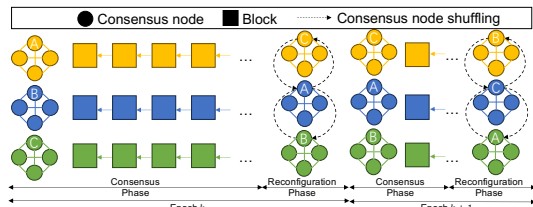

**Figure 1.** The two phases in each epoch of a sharding blockchain. In the consensus phase, transactions in the upcoming blocks are processed. In the reconfiguration phase, the consensus nodes are redistributed for security reasons.

The former phase involves processing transactions and generating blocks, utilizing *state placement* to minimize CSTs. The latter phase occurs every few blocks by shuffling consensus nodes into different shards for security reasons and implementing *state redistribution* to minimize CSTs.

In the consensus phase, the new state is created and placed for the first time. Existing state placement solutions adopted by cutting-edge sharding blockchains [14, 20, 21, 39, 43] for this phase are heuristic in nature, which are simple but less effective in solving the state placement problem. In the reconfiguration phase, the state is redistributed through the reconfiguration of consensus nodes. Based on the rich spatial-temporal (*ST*) characteristics in blockchain transaction data [8, 19, 45], the state redistribution method [15, 24] utilizes community detection and graph partitioning techniques applied to transaction graphs to redistribute states. The ST characteristics of transactions are also valuable in solving the state placement problem as they reveal the possible locations of future states that have transactions with that new state[8, 19, 45]. However, the transaction graph may contain millions of nodes (addresses) and edges (transaction relationships), causing the partitioning process to be time-consuming, taking hundreds of seconds to complete [24]. Consequently, these methods are not feasible in the consensus phase.

To provide an efficient state placement framework in the consensus phase, we propose **Spring**, a new **S**tate **P**lacement framework based on deep **R**einforcement learn**ING** [25] that utilizes the ST characteristics of transactions. As the block can be viewed as a sequence of transactions that contain many new states to be placed, state placement can be viewed as a sequential decision problem suitable for reinforcement learning (RL) [37] approaches. Additionally, since the change of characteristics does not have an explicit pattern as the blockchain evolves [8, 19, 45], RL can adapt dynamically to capture changes that heuristics may fail to grasp. Experimental results based on real Ethereum transaction data show that Spring can solve the state placement problem effectively in different periods of real-world transaction data.

This paper makes the following contributions:

- To the best of our knowledge, Spring is the first to solve the state placement problem using RL. We formulate the Markov Decision Process model for state placement in sharding blockchains, converting the state placement problem into a sequential decision-making task while accounting for computational overhead and the ST characteristics of transactions.
- We propose a sharding blockchain system model and a decentralized agent deployment and training protocol to apply Spring into sharding blockchains.
- We evaluate the effectiveness of Spring on a sharding blockchain testbed deployed on the Alibaba Cloud where the test data set consists of millions of real Ethereum transactions from the years 2015, 2019, and 2023. Spring outperforms SkyChain [44], Monoxide [39], Shard Scheduler [21], and a heuristic baseline in all three time periods. In particular, Spring achieves a throughput improvement of 36.09% and a reduction in the CST ratio of 26.63%. Additionally, Spring strikes a good balance between reducing the CST ratio and maintaining workload balance.

## 2 Background and related work

### 2.1 Deep Reinforcement Learning

RL [37] is a prominent method for addressing sequential decision problems [5]. RL relies on an *agent* to take *actions* continually in an *environment*, receive *rewards* and the new state from the environment based on the outcomes of the actions, and adjust its *policy* to optimize long-term cumulative rewards. Deep Reinforcement Learning (DRL) [25] combines the strengths of deep learning [23] and RL and has been widely applied in sequential decision problems in distributed systems [28, 41].

### 2.2 Sharding Blockchain and Cross-Shard Transactions

Several solutions have been proposed to implement a sharding blockchain [9, 15, 20, 27, 43]. The consensus phase in sharding blockchains includes intra-shard consensus and cross-shard consensus. The former is relatively simple and can be implemented using protocols such as PBFT [7]. The latter, on the other hand, requires interoperability [16] among all shards to handle cross-shard transactions, which is more complex. Common methods for cross-shard transaction processing include the lock-mint two-phase commit mechanism [20], relay-based approaches [24, 39], and appointing one special shard to handle all cross-shard transactions [14]. For the reconfiguration phase, since all consensus nodes are partitioned into smaller committees in different shards, the cost of compromising a shard is lower than compromising the entire blockchain. Therefore, shard reconfiguration is needed to shuffle consensus nodes across all shards to maintain the safety of the entire system.

## 2.3 State Redistribution

BrokerChain [15] is a cross-shard protocol that aims to reduce cross-shard transactions through fine-grained state-graph partitioning and state segmentation mechanisms. BrokerChain collects transaction information during the reconfiguration interval and builds a state graph, which is partitioned using Metis [18]. Subsequently, one state is segmented into multiple sub-states and distributed across shards. To reduce the computation time in BrokerChain, the Constrained Label Propagation Algorithm (CLPA) [24] is proposed for state redistribution. The CLPA is a community detection mechanism that requires less time than Metis.

Overall, state redistribution solutions are storage-space-consuming and time-consuming. However, since these solutions and Spring work in different phases, they are orthogonal and can be combined to reduce cross-shared transactions better. We leave this to future work as we focus on the state placement in the consensus phase.

## 2.4 RL in Sharding Blockchains

SkyChain [44] is the first to apply DRL in sharding blockchains. SkyChain proposes a DRL-based sharding blockchain protocol that enables a sharding blockchain to dynamically change its number of shards, block size, and shard reconfiguration interval at run-time. Clustering-based dynamic sharding (CBDS) [42] focuses on blockchains in the Internet of Things (IoT). CBDS clusters IoT devices using K-means [32] for user grouping and consensus node assignment. CBDS then builds a transaction graph to represent transactions between IoT devices and dynamically adjusts sharding blockchains based on this graph as the state for RL.

**Common limitations**: Currently, research on RL-based sharding blockchain is still in its early stages. There are three main limitations of SkyChain and CBDS: 1) The state space in SkyChain and CBDS does not consider the state of each shard. SkyChain only considers the total number of consensus nodes and pending transactions, failing to reflect the situation at the shard level. 2) The action space design of the agent is inappropriate. In SkyChain and CBDS, the agent's actions include adjusting the reconfiguration interval and the block size. In a blockchain, these parameters are highly related to the consensus algorithm and cannot be arbitrarily modified. For instance, increasing the block size leads to longer transaction propagation time, and may expose the blockchain to security attacks such as double-spending attacks [17]. 3) The dataset used in SkyChain and CBDS is a simulated dataset that does not accurately reflect the ST characteristics of real transactions. Overall, the problem they solve is more like a general scheduling task for a cloud computing platform: when the load (transactions) increases, and there are more resources (consensus nodes), the number of machines (shards) is increased, which does not truly reflect the characteristics of a sharding blockchain. Consequently,

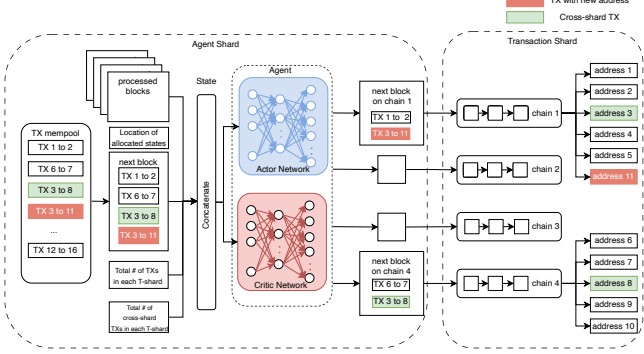

**Figure 2.** SpringChain workflow overview. The client submits transactions (TXs) to the A-Shard, then the leader in the A-Shard selects transactions from the mempool and places new states to T-Shards. The selected transactions are sent to the T-Shards, some of which are cross-shard transactions.

current DRL-based solutions cannot solve the state placement problem. Therefore, we propose Spring to address the limitations of previous methods.

## 3 Sharding Blockchain Design

This section provides an overview of *SpringChain*, the sharding blockchain protocol where Spring can be applied, to show the feasibility of integrating Spring in sharding blockchains.

### 3.1 Basic Design and Assumptions

Similar to prior work [27, 43], we use *epochs* as shown in Fig.1 to represent the term of the consensus nodes. Each epoch spans several consensus rounds for block production. In the reconfiguration phase of each epoch, a verifiable random function [29] is applied to generate unpredictable and bias-resistant randomness, called *Epoch Randomness* (ER). Furthermore, we assume that all nodes have equal computational power.

As blockchain is decentralized, there are *malicious* nodes that stage attacks to compromise the blockchain. Our assumption is that the adversarial parties cannot control more than $f = \frac{1}{3}$ of the consensus nodes and cannot forge signatures. In practice, this is attainable through various Sybil attack prevention mechanisms, such as those employed by well-established blockchains like Bitcoin [30] and Ethereum [40], including proof-of-work (PoW) [30] and proof-of-stake [40]. In SpringChain, we use the PoW to prevent the Sybil attack. PoW requires nodes seeking to join the blockchain to solve a puzzle, and the last few bits of the solution string indicate which shard the node belongs to. All nodes in SpringChain are connected by a partially synchronous [12] peer-to-peer network, where the network may partition, but it will heal after an unknown amount of time.

## 3.2 System Model

Fig.2 provides an overview of SpringChain. As many sharding blockchains [10, 14, 15, 24] have established, SpringChain also consists of two types of shards: *A-shard* and *T-shard*. SpringChain requires one A-Shard and $k$ T-Shards.

- **T-Shard**: The T-Shard refers to the *transaction shard*, which concurrently verifies and processes transactions.
- **A-Shard**: The A-Shard refers to the *agent shard*, which receives users' transaction requests and decides which T-Shard the transaction should be sent to, thus completing state placement.

Similar to the state-of-the-art sharding blockchains [20, 27, 43, 44], both the A-Shard and T-Shard in Spring adopt a Byzantine fault tolerant (BFT)-based consensus protocol, PBFT [7] protocol, as the intra-shard consensus protocol. In addition, a relay-based [24, 39] CST processing model is adopted. In this model, the transaction is first processed on the source blockchain, and then the result is relayed to the target blockchain to finalize the CST.

The workflow of the sharding protocol shown in Fig.2 is described as follows:

1. In each consensus round, the leader in the A-Shard selects $n_t$ transactions from its transaction mempool to produce the state placement result in the form of $k$ transaction batches. After reaching a consensus on the placement result, the batches are sent to the corresponding T-Shards.
2. T-Shards verify and execute the transactions, and only valid transactions are included in the block. Additionally, the new addresses corresponding to the new states and the number of CSTs are recorded in the new block.
3. Finally, by observing the new blocks of T-Shards, consensus nodes in A-Shard record the location of new states, the number of CSTS, and the total number of transactions in each T-Shard for further agent training.

**Overhead of A-Shard.** Although all transactions are first sent to A-Shard, A-Shard will not become a bottleneck. For computing overhead, unlike previous work [9], A-Shard does not actually process transactions, which is the time-consuming part [31]. Transactions are stored in the mempool of the A-Shard node, an in-memory data structure with $O(\log n)$, even constant [11], insertion/deletion time, where $n$ is the number of transactions. For storage overhead, unlike previous work [15, 24] who designates one shard to store the full state of the whole blockchain, A-Shard only stores *address* $\Rightarrow$ *shard* relation to place new states.

Spring assigns the transaction containing the new address to a specific T-Shard to place the new state. When an address appears on the blockchain for the first time, it indicates that a new state containing the metadata, like the balance of the address, is created in the blockchain. For simplicity, assigning the transaction that contains a new address in this

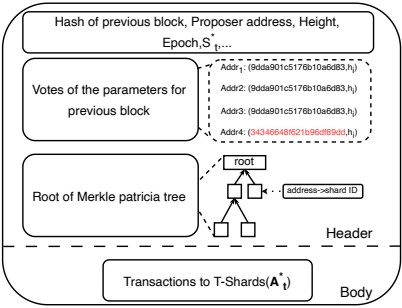

**Figure 3.** The data structure of the block in A-Shard at height $h_{i+1}$. The red text indicates the vote is from a malicious node. The $S_t^*$ and $A_t^*$ are the state of T-Shard and state placement results, respectively, which are used in agent training.

paper actually means placing a new state. Besides, since the meaning of state in RL is different from that in blockchain, to avoid ambiguity, address is used interchangeably to represent state in blockchain in this paper.

## 3.3 Decentralized Agent Deployment and Training in A-Shard

To avoid centralization issues, we describe a decentralized agent deployment and training protocol. Each consensus node in A-Shard maintains a copy of the agent with identical initial parameters. Moreover, the hash of A-Shard's genesis block is used as the randomness seed in all agents. The update of the agent and the state placement actions of all agents are determined via a consensus process. In the consensus phase, one node is selected as the proposer/leader based on ER and proposes a new block.

The data structure of a block in A-Shard is shown in Fig.3, which contains two parts: the *header* and the *body*. The block header includes metadata, votes from peers, and the root of the Merkle Patricia tree (MPT) [40]. The leaf node of the MPT stores the mapping from the address to the ID of T-Shards. With the MPT, the existence and location of the address can be queried efficiently with $O(\log n)$ complexity, where $n$ is the number of addresses.

As all the validity of the content in A-Shard block can be verified via transactions, the basic safety and liveness property of the underlying consensus protocol is not compromised. Consequently, we use PBFT as an example to illustrate how the proposer in A-Shard proposes a block and helps all consensus nodes update the agent consistently. The details of the security analysis of the protocol are in Appendix.A.

**1) Pre-Prepare.** At the start of a consensus round, the leader broadcasts a *pre-prepare* message that contains the proposed block to its peers. As demonstrated in Fig.3, the proposed block includes the state $S_t^*$ of T-Shards that the leader observes in the block header, and the placement results $A_t^*$ based on $S_t^*$ in the block body. $S_t^*$ and $A_t^*$ are RL-related components which will be introduced in Section 4. Upon

receiving the *pre-prepare* message, the consensus nodes will broadcast a *prepare* message which contains the hash of the block to guarantee that all peers received the same block.

**2) Prepare.** Upon receiving *prepare* messages of the same block from more than 2/3 of peers, the consensus node will verify $A_t^*$ from the leader by creating their own $A_t'$ with their local views of state $S_t'$. A predefined threshold $\phi_a$ is set to tolerate the inconsistency of placement results caused by the inconsistent state due to the nature of the distributed system. Since the placement result is actually $k$ transaction batches, the differences between $A_t'$ and $A_t^*$ can be evaluated by the Jaccard index [3]. If the difference is within $\phi_a$, the consensus node broadcasts the *commit* message to confirm the proposed block's validity.

**3) Commit.** The proposer and other nodes accumulate *commit* messages for the proposed block. If more than 2/3 of the consensus nodes send *commit* message for the proposed block, all consensus nodes will commit the block and update the local model. Besides, the leader will send the transaction to the corresponding T-Shard with proof, which indicates the placement result is reached via a consensus round.

## 4 DRL Design

We formulate the state placement problem as a Markov Decision Process (MDP) [33], which is a powerful mathematical framework that captures the essence of sequential decision-making under uncertainty and is the foundation of RL. Specifically, the MDP can be represented as a tuple $(S, A, P, R)$, where:

**State, $S$:** The existing modeling of sharding blockchain has some limitations [15, 24, 42, 44]. Firstly, they do not consider the state of each shard, the state space in Spring is designed to consider the current situation in each shard. Secondly, the state space should not use too much information as graph-partitioning-based solutions do since the size of the transaction graph will continue to grow and increase the overhead of the blockchain node. Thirdly, part of the information in the transaction graph will be outdated due to transactions' changing ST characteristics.

As shown in Fig.2, when a new block arrives, the agent can observe the information of that block, such as the location of senders for each receiver, and the states in all shards, such as the workload distribution and the number of CSTs. Besides, the type of the new address, whether it corresponds to a normal account or a smart contract, is also a factor worth considering, as the transaction characteristics of the two are different [8, 19, 45].

Taking the above-mentioned factors into account, the state $s$ is a $11k + 1$-dimensional vector represented as follows:

$$s = [num\_tx_{11}, \ldots, num\_tx_{5k},$$
$$cross\_tx_{11}, \ldots, cross\_tx_{5k},$$
$$sender\_pos_1, \ldots, sender\_pos_k, flag\ F],$$

where $k$ represents the number of T-Shards. $num\_tx_{wi}$ and $cross\_tx_{wi}$ incorporate the concept of a dynamic sliding window. They refer to the total number of transactions and CSTs in the previous five blocks for the $i$-th T-Shard so that $w \in [1, 5]$ indicates the block index. This sliding window provides the temporal information for the agent because the order of elements in $num\_tx_{wi}$ and $cross\_tx_{wi}$ is set according to the newness of the block. Additionally, $sender\_pos_i$ signifies the position of all senders linked to new addresses within the current block, which reflects the spatial characteristics. The *flag F* is a one-hot indicator with a value of 0 or 1, indicating whether the address is a contract account or an externally owned account.

At the start of each training step, $num\_tx_{wi}$ and $cross\_tx_{wi}$ are initialized based on the recent five processed blocks. Moreover, as the address of senders has been placed in earlier blocks, $sender\_pos_i$ is initialized based on the placement result of existing addresses. The *flag F* is also initialized based on the type of the new address.

Overall, considering the feasibility and decentralization of the blockchain, the state should come from the publicly available content on the blockchain, which limits the range of options. Although subject to this limitation, experimental results show that the information selected from the limited choices can represent the situations in each shard and each new address to an acceptable extent, without introducing a significant storage burden.

**Action, $A$:** Existing RL-based studies select adjusting the block size and reconfiguration interval as the agent's action, which is not applicable in the real sharding blockchain. To overcome this drawback, the agent in Spring opts to directly determine which shard the new address will belong to, which is a more feasible action since it is independent of the parameters of the consensus process. Specifically, the action is an $k$-dimensional one-hot vector, represented as:

$$action = [a_1, a_2, \ldots, a_k], \tag{1}$$

where the value of $a_i$ is either 0 or 1, indicating whether to assign a new address to the $i$-th T-Shard or not. The action space in Spring is concise and directly places the address.

**Transition, $P$:** The state transition function $P$ defines the probability of reaching the new state $s_{t+1}$ after taking action $a_t$ under the given state $s_t$.

$P$ works in the following ways: For each arriving block, the total number of transactions $num\_tx_{wi}$ and the number of cross-shard transactions $cross\_tx_{wi}$ on each T-Shard is only changed at the start of a step according to the situation in T-Shards. For the position of the sender $sender\_pos_i$, it is changed according to each agent's actions within a block. After placing an address to a T-Shard, $sender\_pos_i$ related to this address is updated. The flag is updated based on the type of the current address to be placed.

**Reward, $R$:** The reward is the beacon that guides the agent to achieve our design goal. In reward design, two indices,

i.e., $r_{cstr}$ and $r_{wlb}$, are defined to represent the ratio of CST and the workload balance situation across all shards in the *current block* after taking the action, respectively. The reward function $r$ is defined as follows:

$$r_t = \lambda \cdot r_{cstr} + (1 - \lambda) \cdot r_{wlb} \qquad (2)$$

$$r_{cstr} = \frac{\sum_{i=1}^{k} num\_tx_i}{\sum_{i=1}^{k} cross\_tx_i} \qquad (3)$$

$$avg\_tx = \frac{\sum_{i=1}^{k} num\_tx_i}{k} \qquad (4)$$

$$abs\_diff = \sum_{i=1}^{k} |num\_tx_i - avg\_tx| \qquad (5)$$

$$r_{wlb} = \exp(-\beta \cdot abs\_diff), \qquad (6)$$

where $r_t$ represents the reward at the $t$-th step and $\lambda$ is a weight parameter that balances the importance between the cross-shard transaction ratio and workload balancing. $r_{cstr}$ is the inverse of the ratio of CST as we expect a low ratio of CST. As for $r_{wlb}$, an exponential function of the absolute difference $abs\_diff$ is employed to represent the workload balance situations. To ensure the balance of the workload and easier training, we use the Laplace–Stieltjes transform [6] in $r_{wlb}$. Finally, $\beta$ in $r_{wlb}$ is to control the decay rate.

The overall objective $R$ is to maximize the accumulated reward and can be defined as follows:

$$R = \mathbb{E}_{\tau \sim \pi_\theta} \Big[ \sum_{t=0}^{T} \gamma^t r_t \Big],$$

where $\tau$ is the trajectory that represents an episode of state placement procedure, $\pi_\theta$ is the address policy parameterized by $\theta$, and $T$ is the total number of steps in the trajectory. Each step corresponds to a block. The $\gamma$ is the discount factor. In this paper, Proximal Policy Optimization (PPO) [35] is applied to optimize the agent. The detailed description of PPO is in Appendix.B.

## 5 Evaluation

### 5.1 Experimental Settings

For the evaluation, the A-Shard is implemented via simulation in Python 3.9 and the T-Shard is implemented via a blockchain testbed. The agent is trained on a machine with an Intel(R) Core(TM) i7-9750H CPU @ 2.60 GHz, 64 GB RAM, and an NVIDIA GeForce RTX 2080. Additionally, we verify the trained result with an open-sourced sharding blockchain testbed, BlockEmulator [2], to show the improvement in throughput brought by Spring. BlockEmulator uses PBFT as its intra-shard consensus protocol and a relay-based approach for CSTs, which is consistent with our design described in Section 3. The T-Shard is deployed on 64 ecs.c7.large instances from the same zone in Alibaba Cloud with 16 shards and four consensus nodes in each shard.

To demonstrate the adaptability of Spring, we selected real Ethereum transaction(*TX*) data from 2015, 2019, and 2023 [46], encompassing up to six million TXs. The hyperparameter settings can be found in Appendix.C.

### 5.2 Overhead of SPRING

For the storage overhead, according to our evaluation, the size of an RL model is about 30KB. The computational overhead of Spring is twofold: 1) the cost of assigning an address (*AA*), including the time taken by the agent to take the action of deciding which shard to assign the address to, and 2) the cost of updating the training model (*UTM*). Based on the machine used in this experiment, AA in Spring costs about 0.002 seconds, and UTM costs about 0.1 seconds, which is much lower than the computational overhead of graph-partitioning-based solutions.

Moreover, the computational speed can still be improved. AA and UTM are vector computations, which are related to the computational power of the hardware [22]. Therefore, powerful hardware such as a graphics processing unit can accelerate both UTM and AA. Additionally, some of the addresses are already placed in previous blocks, which means that not all addresses need to be assigned again and therefore less computation time is required. Overall, even with normal computing resources, the computational cost of both network structures of Spring is acceptable.

### 5.3 Exploring Block Size for Improved Performance

Unlike previous DRL-based solutions, the block size in Spring is not adjustable to avoid potential issues. However, the block size is not set arbitrarily. In this experiment, we investigate the impact of block size on the cross-shard transaction ratio (*CSTR*). The block size $n_t$ used in this experiment includes 100, 200, 500, 1000, and 2000. In each experiment, a total of 1000 blocks are used in each experiment to ensure the same amount of training steps. Fig.4 only shows the CSTR with different $n_t$ trained on data from 2023, as results trained with data from other periods are similar. Fig.4 shows the box plot of the cross-shard ratio with different $n_t$. Among them, when $n_t = 1000$, the median value is the lowest. Additionally, the distribution of outliers for $n_t = 1000$ indicates that it has better stability compared to other $n_t$ settings.

Considering that the ST characteristics of TX data can change during runtime [8, 45], $n_t$ can be seen as the *timing window* that represents the ST characteristics within $n_t$ TXs. If $n_t$ is set to be too large, like 2000 in this experiment, the ST characteristics might have changed significantly within $n_t$ blocks. Conversely, a too-small $n_t$, like 100 to 500 in this experiment may not contain enough ST information for effective state placement. Consequently, based on the experimental observations from Fig.4, we set $n_t$ to 1000 in the rest of the experiments.

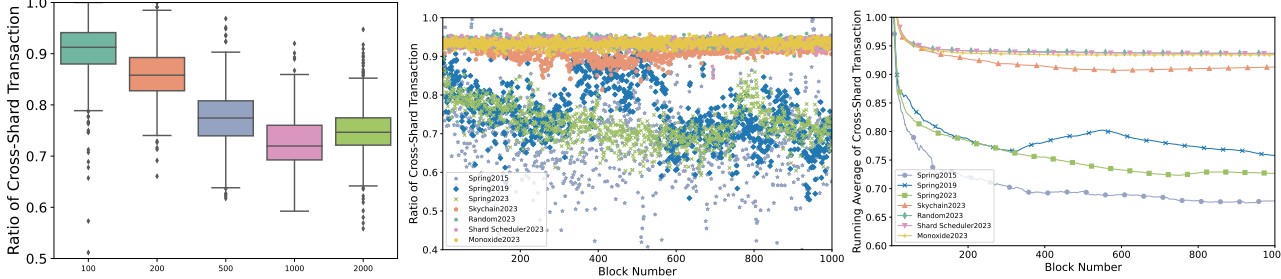

**Figure 4.** Box plot of CSTR for different block sizes with TX data in 2023

**Figure 5.** Scatter plot of CSTR for different algorithms

**Figure 6.** Running average of CSTR for different algorithms

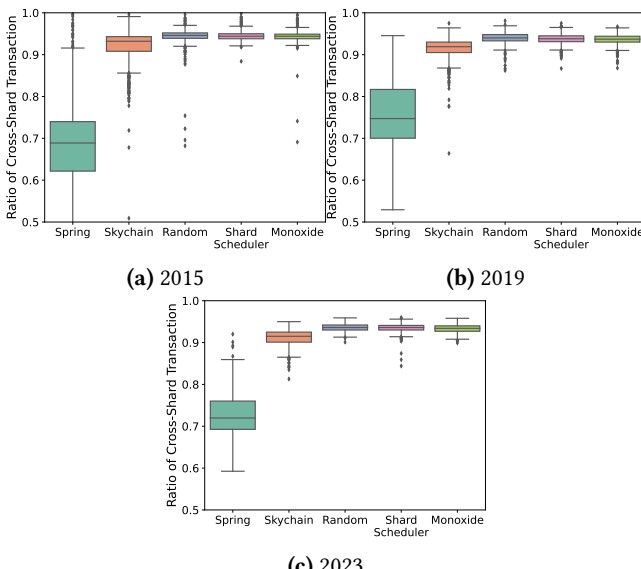

**(a)** 2015

**(b)** 2019

**(c)** 2023

**Figure 7.** Box plot of CSTR for different algorithms with TX data from 2015, 2019 and 2023

## 5.4 Performance Comparison

**Baselines.** We choose the following four baselines to show the effectiveness of Spring:

1. **Random.** Random solutions in previous work [20, 43] randomly select a shard for the new state.

2. **Shard Scheduler** [21]. Shard Scheduler assigns a new state to the shard with the least number of states.

3. **Monoxide** [39]. Monoxide allocates the new state based on the last few hexadecimal digits of the corresponding address. For example, for a new address ending with "e" in a sharding chain system with eight shards, it will be placed in the 7th shard (e % 8 = 6).

4. **SkyChain** [44]. We reproduce SkyChain by using its state and reward for state placement. SkyChain also adopts DRL in sharding blockchains and uses the total number of consensus nodes and pending TXs as its state and the throughput as its reward.

The performance experiments are conducted using a total of one million TXs in each period which corresponds to 1000 blocks as the training data, and various metrics such as CSTR and workload distribution are collected during the training process. These experiments aim to demonstrate the effectiveness of Spring compared to the chosen baselines.

### 5.4.1 Cross-shard Transaction Ratio.
Fig.7a to Fig.7c illustrates the CSTR using different state placement algorithms. For all algorithms, except Spring and SkyChain, the CSTR is at around 94% in all periods, showing the effectiveness of DRL. Moreover, Spring significantly outperforms all other baselines in all periods. As Spring takes advantage of the ST characteristics of the TX data, it can make more judicious state placement decisions. Overall, Spring achieves up to 26.63% reduction in CSTR compared to other baselines.

Fig.5 and Fig.6 take a deeper view into the training process. As the baselines have similar performance, we only show the performance of baselines with data from 2023. Fig.5 shows that in most of the training steps Spring outperforms all the baselines in all periods. Besides, Fig.6 demonstrates that Spring can learn the ST characteristics to reduce CSTR efficiently, since we train the agent from scratch with data in all three periods. It is worth noting that, in the middle of the training with 2019 data, the CSTR rises. This could be the case of the change in the ST characteristics. For example, most TXs are issued from or to several addresses, with the counterpart addresses spread out across different shards. After this period, the CSTR decreases, showing Spring possesses good adaptability.

For the performance of each algorithm, since SkyChain uses throughput as a reward, which is related to CSTR, it can also find a way to reduce CSTR to some extent. However, it does not consider the state within each shard and the location of the sender address associated with the receiver, so its actions are taken based on the less informative state compared to Spring, resulting in a limited reduction in CSTR. The heuristic algorithm also has the same issue of not being able to allocate the state to an appropriate shard.

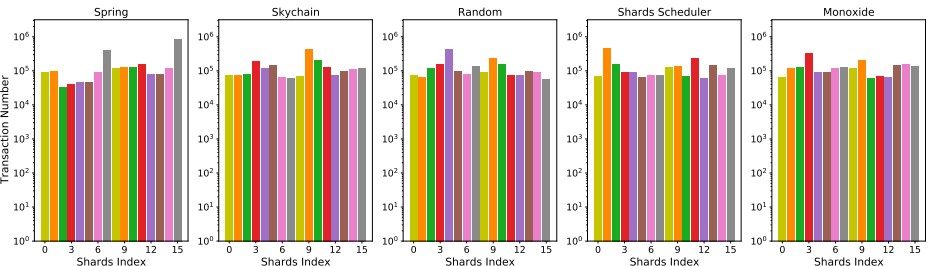

**Figure 8.** TX distribution of all shards in different algorithms

Overall, Spring has good adaptability and successfully utilizes a DRL-based state placement approach that leverages the ST characteristics of TX data to reduce the CSTR.

**5.4.2 Load Balancing Exploration.** In this part, the load-balancing situation in five state placement strategies is analyzed. Fig.8 shows the accumulated TX distribution in different state placement strategies for the 1000 blocks. The y-axis of Fig.8 is in log-scale. Overall, there is a trade-off between the cross-shard ratio and workload balance. Previous work such as Monoxide does not guarantee optimization of both factors [15]. However, Spring reduces the cross-shard ratio without excessively sacrificing workload balance.

It can be observed from Fig.8 that all strategies suffer from some level of workload imbalance. The imbalance in all algorithms is attributed to the power law distribution of TXs, which has been observed in Ethereum TX data analysis[8, 45]. This indicates that a small fraction of addresses are responsible for a large number of TXs. Taking the situation in 2015 as an example, 9 out of 33,899 addresses contribute to 73.7% of TXs, following the power law distribution. The TX distribution also follows the power law in other periods. These popular addresses are usually smart contracts or mine pools[8, 45]. Moreover, since Fig.8 represents the cumulative result, the difference between shards is smaller when a block contains only 1000 TXs, meaning that the influence of the imbalance could be amortized across all shards.

Furthermore, it should be emphasized that balance in address distribution does not guarantee balance in TX distribution. For instance, although the Shard Scheduler evenly places addresses across shards, the distribution of TXs is not even as shown in Fig.8. This holds true for Spring as well. For instance, although there are about 25% more addresses in T-Shard 7 (index starts from 0) compared to T-Shard 6, the difference in the number of TXs between them is only 11%. Additionally, the total number of addresses in 2019 and 2023 is greater than in 2015. This is because, as users become more concerned about privacy, mixing services [36] have been widely used since 2019, which generates many one-off addresses. The placement result of Spring is consistent with the findings from Ethereum TX data analysis[8, 45], indicating that Spring successfully captures the characteristics of the TX data.

**Table 1.** TPS of different algorithms

| Algorithm | TPS | Algorithm | TPS |
|-----------|-----|-----------|-----|
| Spring2015 | 486.8532 | Monoxide2023 | 325.542 |
| Spring2019 | 426.424 | ShardScheduler2023 | 321.957 |
| Spring2023 | 437.961 | Random2023 | 326.691 |
| | | SkyChain2023 | 342.35 |

Moreover, the results could potentially be further improved if state redistribution is considered during the reconfiguration phase, which could be explored in future work.

**5.4.3 Throughput Exploration.** The throughput is defined as the number of TXs the sharding blockchain can process per second (TPS). We collect the throughput using BlockEmulator [2] to process the TXs in different periods. We modify the state placement module of the BlockEmulator to verify the effectiveness of Spring. As shown in Table.1, Spring consistently outperforms the other algorithms. The performance of baselines is similar in all periods. Thus, we only present the result in 2023. Moreover, the TPS is improved up to 36.03% with 2023 TX data, which is higher than the reduction in CSTR, which is 20.99%. This indicates that CST's impact on the TPS is significant. Since the cross-shard needs an extra processing mechanism, it is more time-consuming. Consequently, reducing the CSTR does improve throughput, and Spring outperforms other baselines.

## 6 Conclusion

In this paper, we present Spring, a deep reinforcement learning (DRL)-based state placement method that first models the Markov Decision Process for state placement. Our solution minimizes the proportion of cross-shard transactions without unduly compromising the workload balance between different shards. In addition, our approach exploits the spatial-temporal properties of transaction data, resulting in improved overall system throughput. Compared to traditional graph partitioning methods, Spring has lower computational and storage overheads. In future work, Spring can be combined with a reconfiguration mechanism to improve the performance of sharding blockchains.

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

## A Security Analysis

**Shard member selection.** In the reconfiguration phase, all shards are reconfigured to rotate the consensus nodes within them, ensuring that the failure probability of all shards remains lower than the safety threshold. Since an adversary can accumulate in one shard to gain control over it, the failure probability of A-Shard or T-Shard in Spring is calculated using the hypergeometric distribution [13]:

$$Pr[X \geq \lceil f n_s \rceil] = \sum_{x=\lceil f n_s \rceil}^{n_s} \frac{\binom{fn}{x}\binom{n-fn}{n_s-x}}{\binom{n}{n_s}},$$

where $n$ denotes the total number of consensus nodes, $n_s$ is the number of consensus nodes in a shard, and $f$ is the fraction of malicious nodes, which is $\frac{1}{3}$ in this paper as A-Shard and T-Shard adopt PBFT. With a sufficient number of consensus nodes, the value of $Pr[X \geq \lceil f n_s \rceil]$ can be lowered to as low as $10^{-6}$.

**Safety and liveness.** As T-Shard only processes transactions like a traditional blockchain with PBFT as its consensus protocol, we mainly focus on the *safety* and *liveness* of the protocol mentioned in A-Shard. The safety property guarantees malicious nodes will not compromise the blockchain, and the liveness property indicates all consensus nodes will finally reach a consensus on the proposed block. We use $\frac{1}{3}$ as the safety threshold since we assume that the number of malicious nodes is less than $\frac{1}{3}$ of the total consensus nodes in Section.3.

**Theorem 1.** *A-Shard achieves safety if there are less than 1/3 of the nodes in the A-Shard are malicious.*

*Proof.* Assuming the leader is malicious and sends arbitrary state placement result $A_t^*$ and its view of state $S_t^*$ to other nodes. According to the *Prepare* phase, all nodes will verify the validity independently. Consequently, a malicious leader cannot compromise the A-Shard. Moreover, Assuming some

consensus nodes are also malicious, they can only vote for the malicious leader, as shown in the red font in Fig.3, or refuse to vote for the honest leader. However, as the malicious nodes are less than 1/3, and the block is valid only when more than 2/3 of the consensus nodes vote for it, malicious nodes cannot make an invalid block valid. Overall, safety can be ensured in A-Shard.

**Theorem 2.** *Spring achieves liveness if there are less than 1/3 of the nodes in the A-Shard are malicious.*

*Proof.* Assuming a partially synchronous network, the proposed block will eventually reach the honest nodes. If a valid block is not produced due to malicious nodes crashing or misbehaving during the consensus round, the consensus round will time out and switch to the next one. The leader will be evicted from the peer connection table of other consensus nodes. Overall, liveness can be ensured in A-Shard.

## B Agent Network Structure and Update Algorithm

We choose a feed-forward neural network with three linear layers and two ReLU [4] activation functions in the agent neural network design. The input layer contains $in_D$ neurons. The two hidden linear layers both have a $n_{neuron}$-dimensional feature space, and the output layer maps the $n_{neuron}$ neurons to $out_D$ neurons. ReLU activation function is used between each linear layer.

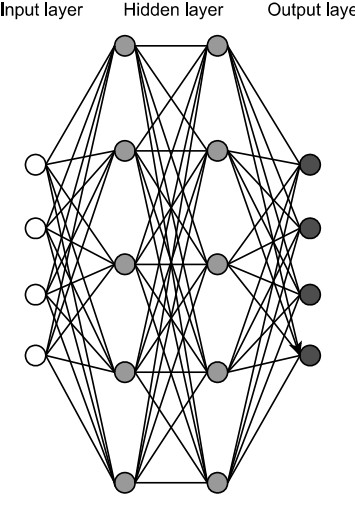

**Figure 9.** Schematic diagrams of agent network structures

1. Input layer: This layer has a dimensionality of $in_D$, which is the state observation of the agent.
2. Hidden layer 1: A fully connected layer, followed by a ReLU activation function.
3. Hidden layer 2: Another fully connected layer, also followed by a ReLU activation function.

4. Output layer: A fully connected layer with a dimensionality equal to the action space dimension ($out_D$) for the actor-network, or a single output neuron for the critic network.

As for deep reinforcement learning, a *policy* is a mapping from states to actions, denoted by $\pi(a|s)$, which represents the probability of taking action $a$ in state $s$. It can be optimized to maximize the expected cumulative reward:

$$\pi^* = \arg\max_{\pi} \mathbb{E}\Big[ \sum_{t=0}^{T} \gamma^t R(s_t, a_t, s_{t+1})\big|\pi \Big].$$

Here, we adopt the PPO algorithm to optimize the address assignment policy. PPO consists of an *actor network* and a *critic network*, which are built using a neural network and are responsible for making action decisions and estimating the value of the action, respectively.

During an episode, the agent interacts with the environment using the existing policy $\pi_\theta$ (actor-network) to collect a batch of data. Once a complete batch of data is obtained, the actor-network and critic-network learn from the sampled data, following the algorithm 1. Specifically, the actor network is optimized using surrogated Policy Gradient as follows:

$$L(\theta) = \mathbb{E}_t\big[\min(ratio_t A_t, clip(ratio_t, 1 - \epsilon, 1 + \epsilon)A_t)\big],$$
$$ratio = \mathbb{E}_t\big[\exp\big(\log(\pi_\theta) - \log(\pi_{\theta_{old}})\big)\big],$$
$$A_t = \delta_t + (\gamma \cdot \lambda) \cdot \delta_{t+1} + \cdots + (\gamma \cdot \lambda)^{T-t} \cdot \delta_T,$$
$$\delta_t = r_t + \gamma * V(s_{t+1}) - V(s_t),$$

where $A_t$ is the Generalized Advantage Estimation [34], which measures the relative advantage of taking action $a_t$ in state $s_t$ compared to the average situation. The critic network can be optimized with Temporal Difference [38] (TD):

$$L_{critic} = \mathbb{E}t\big[(V(s_t) - V_{target}(s_t))^2\big],$$
$$V(s_t) = \mathbb{E}_{a_t}\big[r_t + \gamma * \mathbb{E}_{s_{t+1}}[V(s_{t+1})]\big].$$

By utilizing the above objective functions, the parameters of the actor network and critic network can be continuously updated over multiple episodes, leading to improved performance of the agent in the environment.

## C  Hyper-parameters Settings

**Table 2.** Hyperparameters and their values

| Hyperparameter | Value |
|---|---|
| DRL training batch, $batch$ | 2048 |
| Learning rate, $lr$ | 3e-4 |
| Clip value in PPO, $clip$ | 0.2 |
| Discount factor, $\gamma$ | 0.99 |
| Decay rate weight factor, $\beta$ | 0.1 |
| Weight factor, $\lambda$ | 0.5 |
| number of neurons in each layer, $n_{neuron}$ | 64 |

---

**Algorithm 1** PPO Overview

1: Initialize policy network parameters $\theta$ and critic network parameters $w$.
2: Collect a set of trajectories $D = \tau$ by running the current policy $\pi_\theta$ in the environment.
3: **for** each state-action pair $(s_t, a_t)$ in $D$ **do**
4:     Calculate the target value function $V_{target}(s_t)$
5:     Compute the advantage function $A_t$
6: **end for**
7: **for** K iterations **do**
8:     Perform optimization on the policy network using Adam with mini-batches sampled from $D$, computing the gradient $\hat{g}$ by PPO surrogate objective $L(\theta)$.
9:     Optimize the critic network B times by minimizing the squared TD error $L_{critic}$. Update critic network parameters $w$ using Adam, and compute the gradient $\hat{g}_{critic}$.
10: **end for**

---

The hyperparameter settings used in Spring are summarized in Table 2. We set the batch size for model training to $batch = 2048$. After comparing surrogate objectives, we find that the best learning effect on the environment policy is achieved when the clip value is set to 0.2. For the Adam optimizer, we set the initial learning rate to $lr = 3 \times 10^{-4}$. The system parameters like transaction size $S_T$ and block header size $S_H$ are set according to real blockchain systems like Ethereum [40] or other related work like SkyChain [44].