# OpenReview forum: "SPRING: Improving the Throughput of Sharding Blockchain via Deep Reinforcement Learning Based State Placement"
_ACM.org/TheWebConf/2024/Conference — TheWebConf24 Oral_

### Official Review · Reviewer_TgcR · 2023-11-06

**Novelty:** 5
**Technical Quality:** 5

**Review:**

### Summary:
The paper looks into the challenge of reducing the overhead of cross-shard transactions by applying specific modeling to better exploit reinforcement learning-based coordination approaches. By tapping into spatial-temporal information, the proposed approach SPRING tries to address the weakness (unutilized potential) of other approaches in the area. The paper also features an extensive evaluation, which also incorporates real-world data, to detail how SPRING outperforms related approaches.

### Pros:
+1: The evaluation section is quite extensive and considers related work as well as real-world transaction data.

+2: The paper is well written and nicely presents the approach and findings.

### Cons:
-1: The difference between SPRING and SpringChain is not clearly presented in the paper.

-2: Certain aspects of the evaluation could be improved to further strengthen the paper's contributions.

In my view, the paper is in good shape, which allows the reader to easily follow the presentation, the conducted experiments, and the drawn conclusions. The ratio of CSTs is still a lot, but it appears to be the best approach that we have these days. I also like how the authors manage to implicitly embed motivation and research gap in Sections 2.2 and 2.4. As I will detail below, from my point of view, a few minor issues impair the paper's quality, but nothing too critical.

### Detailed Comments:

#### -1: SPRING vs. SpringChain
In the current version, the term "SpringChain" simply pops up at the beginning of Section 3. Since the paper has no paper organization and the relation between SPRING and SpringChain is not discussed either, this aspect introduces a little bit of confusion. I would like the authors to make the differences, why they are needed, and what implications this separation has more explicit. From my understanding, the separation is mostly between Sections 3 and 4. Is this conclusion correct?

Finally, the writing style of SPRING is inconsistent. Sometimes, it is capitalized, but most of the time, it is not. To add a certain recognition value, I recommend the authors to always use the capitalized form. Moreover, this change can help to better separate it from the term/concept SpringChain.


#### -2: Evaluation
While the evaluation is already quite extensive, and most aspects are well argued, also in terms of which evaluations have been conducted and how the evaluation parameters have been chosen. Regardless, a few minor improvements remain, in my opinion.

First, I would like to know more about the hyperparameter settings that have been selected and whether they are universally applicable. This information is only briefly discussed in the appendix.

Second, the overhead evaluation/discussion is rather brief. To what do the individual processing times accumulate in real-world settings? What is the frequency of these actions/procession steps? Additionally, the paper could better stress which parties are affected by the respective overhead. This information is not given at the moment. I believe that being more precise in this part of the evaluation would improve the paper.

Third, I am not able to follow how the real transaction data, or more precisely, the addresses, are split across shards. To my understanding, this behavior is not directly related to the state, is it? I doubt that this will have a significant impact on the results, but I would still like to know how the authors deal with this aspect.

Finally, the performance of SPRING is not compared to CBDS, which has also been introduced in the related work section (jointly with SkyChain). Without a closer look, I cannot find a reason for this decision. Why did the authors decide to not compare SPRING to CBDS?

#### Other:
- The current presentation has little pointers to the web. Hence, the paper could stress more explicitly why it is relevant for the conference's and track's topics.
- In Section 3.3., why are the proposer and other nodes highlighted as part of the commit step? Is my assumption that this steps covers all nodes incorrect?
- I am a little bit surprised that most of the baselines (especially Monoxide) have not been introduced in the related work section. Why has Monoxide not been presented in the related work section?
- The side note on 9 addresses being responsible for 74% of the transactions is interesting. Wow, I did not know that.


### Nits:
- Figure 1: The white text on yellow color is hard to read, especially on a printout.
- Section 3.2: The writing style of A-Shard and T-Shard is inconsistent.
- Section 3.2: Is "CSTS" correct, or should the last "s" rather be lowercase?
- Section 3.2: "who designates" should probably be "which designates".
- Section 4: "Appendix.B" has a period, which should be removed.
- Section 5: The spacing in "Fig.X" seems to be missing repeatedly.
- Section 5.3: Adding a reference to the initial statement of this section or at least a pointer to the related work section could be beneficial.
- Figures 4-7: The figure sorting is not in order of appearance, but I also have no better idea at this point how to address this nit.
- Figure 8: I suggest including the year of the underlying evaluation data in the figure caption as well.
- Section 5.4: "Table.1" has a period, which should be removed.
- Quite a few times throughout the paper, a space before the citation marker is missing, e.g., "analysis[8,45]".

### Post-Rebuttal

I kindly thank the authors for responding to the reviews.
While the response helps to clarify a lot of aspects, ~~it fails to convey whether changes will be made to the original manuscript~~ (except for my query on SPRING vs. SpringChain).
This approach makes it challenging to assess whether a revised version could convincingly resolve all presentation issues of the paper.
Moreover, the situation amplifies when considering the breadth of the feedback raised by all reviewers.

*Update:* Thank you for providing more details on your revision plan, which is helpful to estimate the planned changes.
Unfortunately, the plan only lists a few omissions, making it challenging to really incorporate all changes in a convincing way within the page limits.
Personally, I believe that the paper would benefit from another round of reviews once the outlined changes have been made.

**Questions:**

What is the reason for not considering CBDS as a baseline in the evaluation?

**Reviewer Confidence:**

3: The reviewer is confident but not certain that the evaluation is correct

**Scope:**

2: The connection to the Web is incidental, e.g., use of Web data or API

---

### Official Review · Reviewer_f3Ry · 2023-11-14

**Novelty:** 3
**Technical Quality:** 4

**Review:**

**Paper summary**

This paper targets the scalability issue in the state placement in sharding blockchains. It develops Spring, which takes the cross-shard transaction ratio and workload balancing into consideration, and then uses deep reinforcement learning to infer state placement policy. SPRING is applied in sharding blockchains, and evaluated using real Ethereum transaction data in 2015, 2019, and 2023. Experimental results demonstrate that it performs better than four baselines in terms of throughput and reduction of CST ratio.

**Strengths**

+ The formulation of the state placement problem as an MDP is a reasonable modeling choice, aligning with the sequential nature of blockchain transactions.

+ The paper uses real Ethereum transaction data for experimental evaluation.

+ The paper is well-structured.

**Weaknesses**

- Novelty and new technical contributions seems lacking. Given the solution of SkyChain, it is a bit difficulty to justify the novelty and the new technical contributions provided by this paper.

- Some technical details about the DRL process should be made clear in the experimental settings, such as the model architecture, training parameters, and hyperparameter tuning.

**Detailed comments**

Spring proposes to use DRL to address the problem of state placement in sharding blockchains. It formulates the problem as an MDP, so that DRL can well handle the sequential nature of blockchain transactions. Below I mainly elaborate on the weaknesses listed above.

*Novelty and innovations*

Despite the common limitations discussed in Section 2.4, the novelty and new technical contributions of Spring haven't been inadequately justified, especially when compared to the existing solution SkyChain. I suggest the paper include a detailed and explicit discussion on the unique technical aspects that differentiate Spring from SkyChain. In the current writing, it is a bit challenging to identify the innovation and inspiring contributions of Spring.

*Technical details*

The paper should also provide more technical details regarding the architecture of the DRL model, training methodologies, and hyperparameter tuning. This will enhance its reproducibility.

How is Ethereum transaction data embedded with BlockEmulator? Or is it just used for training?

There should be a benchmarking on the reproduced SkyChain, to ensure the fidelity of its re-implementation.

Figure 5 and 6 show that Spring's performance in the data of 2015, 2019 and 2023 differ. What make these differences?

**Questions:**

See my review above

**Reviewer Confidence:**

3: The reviewer is confident but not certain that the evaluation is correct

**Scope:**

3: The work is somewhat relevant to the Web and to the track, and is of narrow interest to a sub-community

---

### Official Review · Reviewer_sqUb · 2023-11-19

**Novelty:** 4
**Technical Quality:** 4

**Review:**

The authors present Spring, a deep-reinforcement-learning (DRL)-based sharding framework for state placement. Spring formulates the state placement as a Markov Decision Process which takes into consideration the cross-shard transaction ratio and workload balancing, and employs DRL to learn the effective state placement policy.

Pros:
1. This paper proposes a practical method.
2. The authors conducted detailed experiments.

cons:
1. This paper only discusses two related papers that use RL in sharding blockchains.This paper lacks discussion and comparison of some important papers, such as "DQN-Based Optimization Framework for Secure Sharded Blockchain Systems"，"Sharding for Blockchain based Mobile Edge Computing System: A Deep Reinforcement Learning Approach".

2. In Section 5.2, the authors claim that "updating the training model (UTM) costs about 0.1 seconds". It is necessary to explain under which experimental parameters this result is obtained.

3. Some minor issues:
	a) The font in the figures is too small.
	b) Insufficient description of the dataset in Section 5.1.

**Questions:**

1. Discussion and comparison of more relevant papers.

**Reviewer Confidence:**

4: The reviewer is certain that the evaluation is correct and very familiar with the relevant literature

**Scope:**

3: The work is somewhat relevant to the Web and to the track, and is of narrow interest to a sub-community

---

### Official Review · Reviewer_gtQj · 2023-11-21

**Novelty:** 3
**Technical Quality:** 4

**Review:**

This article proposes Spring, a blockchain sharding framework based on deep reinforcement learning. This paper models the state placement of sharding blockchain as a Markov model and provides a solution to reduce cross-shard transactions using deep reinforcement learning. Evaluation is conducted on the historical dataset of Ethereum. The results shows that Spring can reduce cross-shard transaction ratio by about 26%, with a small calculating overhead.

Pros：
+ The state placement problem of sharding blockchain is a very interesting topic that can have a significant impact on the actual operation of blockchain.
+ Compared to the work of SkyChain and others, Spring's experiments are based on real Ethereum historical datasets.
+ Writing is good. The author's writing looks very smooth.

Cons：
+ While I believe there are some novel features in Spring, at the end of reading the paper I am not completely sure it surpasses existing literature (i.e. SkyChain, mostly) in a way that warrants a top conference publication.
+ Compared to SkyChain, Spring considers load balancing and ST characteristics between different shards when establishing a reward mechanism for reinforcement learning. Please explain why Spring has made significant improvements in CSTR compared to SkyChain in Figure 7. Is it because Spring utilizes historical data? Is it because there are many duplicate input data in Ethereum's real data?
+ The authors also mentioned in the article that there is a trade-off between the cross-shard ratio and workload balance. According to Figure 8, Spring's load balancing seems to be inferior to SkyChain. I think Spring sacrifices workload balance to improve CSTR. Both of these seem to have an impact on throughput. So, why does Spring with poor load balance and high CSTR outperform other solutions in terms of throughput?
+ Spring uses deep reinforcement learning for state placement, but the text does not seem to mention where reinforcement learning should run. If running on a blockchain using smart contracts, does the smart contract support the cost of deep reinforcement learning? As far as I know, Ethereum smart contracts have gas limits that seem insufficient to support retraining for deep reinforcement learning. If it is executed offline, then reinforcement learning is executed on each agent node, and then consensus is required?
+ The author introduced λin Reward to balance the impact of workload balance and CSTR on Reward. What impact will this hyperparameter have on throughput and CSTR? Please explain how the hyperparameter is set.
+ The font of the image is too small (such as the legends in Figures 5 and 6), and it will definitely not be clear when printed.

**Questions:**

+Please explain the impact of load balancing and CSTR on throughput.
+Please explain why Spring outperforms other solutions in terms of CSTR metrics.
+Please explain the operational location of deep reinforcement learning.
+Please explain the impact of lambda selection in Reward on throughput.

**Reviewer Confidence:**

3: The reviewer is confident but not certain that the evaluation is correct

**Scope:**

3: The work is somewhat relevant to the Web and to the track, and is of narrow interest to a sub-community

---

### Official Review · Reviewer_Vhz8 · 2023-11-22

**Novelty:** 5
**Technical Quality:** 5

**Review:**

The paper at hand proposes a protocol to improve the transaction throughput of sharding blockchains by systematically reducing the number of costly transactions between shards using deep reinforcement learning.
Namely, a network of agent nodes maintains a model for assigning newly observed addresses to a shard based on the blockchain's recent history.

Overall, the authors' goal is intuitive and the chosen approach seems sensible, but the paper raises some important but unanswered questions regarding the underlying (effective) scenario and some of the design choices (see Questions section).
Furthermore, I believe that an extension of the relevant background on deep reinforcement learning and previous approaches, while keeping the information condensed, would help a broader audience to better understand the concepts and approaches.
Similarly, the authors should ensure that the notation and terminology is used consistently and easy to follow.

Crucially, the deep-learning approach presented in Section 4 seems to have notable flaws as far as I am concerned:

- The components of the reward function $r_{t}$, $r_{cstr}$ and $r_{wlb}$, have different effective domains that the weighing parameter $\lambda$ cannot catch. Namely, $r_{cstr}$ can grow (theoretically) infinity, whereas $r_{wlb}$ lies between 1.0 and 0.0.
- In a performance-wise ideal scenario, there would be no CSTs at all, leading to a hypothetical division by zero for $r_{cstr}$.
- In fact, the authors assume a low ratio of CSTs (increasing $r_{cstr}$). However, this assumption stands in stark contrast to their observation in Section 5.4.1, indicating that current approaches suffer from approximately 94% of all transactions being CSTs.

*Suggested improvements:*

- The introduction already dives into a detailed technical discussion of sharding blockchains. I suggest keeping this discussion as concise as possible to still support the motivation, but move details to Section 2.2 to have all background information in one place.
- Make more clear what the *state* is in a sharding blockchain, where it is located, and what implications shuffling nodes between committees has on transferring the required state
    - Further, please make the distinction between the state of the RL-model and the blockchain's state (addresses of users and smart contracts) as early as possible and ensure to avoid ambiguities. In the presented manuscript, this aspect especially is a source for confusion.
- Section 3.2 already discusses performance overheads. I suggest limiting this aspect to a mere intuition at this point and move the detailed discussion to Section 5.
- Section 4 relies on unintuitive notation, as $num\textunderscore tx_{11}$, for example, could be misinterpreted. Possible alternatives are $num\textunderscore tx_{1,1}$ or $num\textunderscore tx_{1}^{i}$.

*Minor issues:*

- Spaces are sometimes missing when using references or citations, or introducing an acronym.
- There is an inconsistency in capitalizing Spring/SPRING in the title and body of the paper.
- In Section 2.3, the layout of the state graph used by BrokerChain remains unclear without referring to [15, Section III-B], and it is also unclear whether these details are needed going forward; however, this issue could also relate to the ambiguous use of "state" (see above).
- In Section 3.1, $f = 1/3$ should read $f < 1/3$.
- Minor inconsistency: In Section 3.3, PBFT is being used as an "example," but Section 3.2 already fixed the usage of PBFT as a design choice.
    - Furthermore, it is only implicit that the pre-prepare, prepare, etc. messages are part of PBFT and not Spring.
- In Section 5.3, there seems to be an artefact of an older version of the sentence referencing Figure 4.

Update: I acknowledge that I have read the authors' rebuttal comments.

**Questions:**

**Scenario-related:**

- What are the implicit underlying payment patterns assumed? While Section 3.1 mentions some assumptions, I have two unaddressed concerns/questions:
    1. How stable must the transaction flows be to even have a chance to build even a short-term model in the general case? I assume that stable, reoccurring patterns lend themselves to the authors' approach, but I would expect more seemingly random, one-off payments to decrease the effectiveness of Spring.
    2. Similarly, what is the *impact* of heavy hitters, i.e., few addresses that occur in many transactions, such as exchange services? Section 5.4.2 discusses that heavy hitters indeed exist, but does that not imply that such heavy hitters become more likely to be involved with transactions from every shard regardless? This question probably boils down to the following: Of the 94% reported cross-shard transactions, what is the theoretical baseline for achievable reduction given the presence of heavy hitters?
- Regarding Section 5.4: Isn't it to be expected that Random, Shard Scheduler, and Monoxide behave very similarly with respect to CSTs? None of the strategies takes the payment flows into account and thus I would have expected that they all show random behavior in these experiments.

**Design choices:**

- I had a hard time grasping the DRL state layout proposed in Section 4:
    - Does $s$ cover a global state or does it encode a single update based on one transaction? The overall state layout seems to imply the former, but then I do not understand how the flag $f$ works.
    - What is the intuition behind $sender\_pos_{i}$? I did not understand the provided explanation.

**Ethics Review Description:**

-

**Reviewer Confidence:**

2: The reviewer is willing to defend the evaluation, but it is likely that the reviewer did not understand parts of the paper

**Scope:**

3: The work is somewhat relevant to the Web and to the track, and is of narrow interest to a sub-community

---

### Decision · Program_Chairs · 2024-01-22

**Decision:**

Accept (Oral)

**Comment:**

The paper received 5 reviews. One was negative leaning; the others were positive leaning. The authors engaged with the reviewers during the rebuttal phase and addressed several concerns. I received a final recommendation from 3 reviewers, which was borderline, weak, accept and accept. Based on these recommendations, the reviews and the discussions with the authors, I recommend a weak accept.

 The paper has strong merits, but a few issues prevent it from making an unequivocal positive recommendation. The main issue is the technical novelty given the existence of the referenced SkyChain work and clarity around the DRL approach and how it fits within the framework.